# LEARNING DEEP GENERATIVE MODELS WITH DISCRETE LATENT VARIABLES

## ABSTRACT

There have been numerous recent advancements on learning deep generative models with latent variables thanks to the reparameterization trick that allows to train deep directed models effectively. However, since reparameterization trick only works on continuous variables, deep generative models with discrete latent variables still remain hard to train and perform considerably worse than their continuous counterparts. In this paper, we attempt to shrink this gap by introducing a new architecture and its learning procedure. We develop a hybrid generative model with binary latent variables that consists of an undirected graphical model and a deep neural network. We propose an efficient two-stage pretraining and training procedure that is crucial for learning these models. Experiments on binarized digits and images of natural scenes demonstrate that our model achieves close to the state-of-the-art performance in terms of density estimation and is capable of generating coherent images of natural scenes.

## 1 INTRODUCTION

Building generative models that are capable of learning flexible distributions over high-dimensional sensory input, such as images of natural scenes, is one of the fundamental problems in unsupervised learning. Historically, many multi-layer generative models, including sigmoid belief networks (SBNs) (Neal, 1992), deep belief networks (DBNs) (Hinton et al., 2006), and deep Boltzmann machines (DBMs) (Salakhutdinov & Hinton, 2009), contain multiple layers of binary stochastic variables. However, since the debut of variational autoencoder (VAE) (Kingma & Welling, 2013) and reparameterization trick, models with continuous variables have largely replaced previous discrete versions. Many improvements (Burda et al., 2015; Kingma et al., 2016; Chen et al., 2016; Gulrajani et al., 2016) along this direction have been pushing forward the state-of-the-art for years.

Comparing with continuous models, existing discrete models have two major disadvantages. First, models with continuous latent variables are easier to optimize due to the reparameterization trick. Second, every layer in models, including SBNs and DBNs, is stochastic. Such design pattern restricts the depth of these models because adding one layer can only provide small additional representation power while the extra stochasticity increases the optimization difficulty and thus out-weights the benefit.

In this paper we explore learning discrete latent variable models that perform equally well with its continuous counterparts. Specifically, we propose an architecture that resembles the DBN but uses deep deterministic neural networks for inference and generative networks. From the VAE perspective, this can also be seen as deep VAE with one set of binary latent variables and learnable restricted Boltzmann machine (RBM) prior (Hinton, 2002). We develop a two-stage pretraining, training procedure for learning such models and show that they are necessary and effective. Finally, we demonstrate that our models can closely match the state-of-the-art continuous models on MNIST in terms of log-likelihood and are capable of generating coherent images of natural scenes.

## 2 BACKGROUND

Although discrete models are largely replaced by continuous models in practice, there has been a surge in the interest of the learning algorithms that accommodate discrete latent variable models,

such as sigmoid belief networks (SBNs) (Mnih & Gregor, 2014; Bornschein & Bengio, 2014; Mnih & Rezende, 2016). In this section, we briefly review those methods that lay the foundation of our learning procedure.

To learn a generative model $p(\mathbf{x})$ on a given dataset, we introduce latent variable $\mathbf{z}$ and decompose the objective as $\log p(\mathbf{x}) = \log \sum_{\mathbf{z}} p(\mathbf{x}, \mathbf{z})$. Posteriors samples from $p(\mathbf{z}|\mathbf{x})$ are required to efficiently estimate the exponential sum $\sum_{\mathbf{z}} p(\mathbf{x}, \mathbf{z})$. However, when $p(\mathbf{x}, \mathbf{z})$ is parameterized by a deep neural network, exact posterior sampling is no longer possible. One way to overcome it is to simultaneously train an inference network $q(\mathbf{z}|\mathbf{x})$ that approximates the true posterior $p(\mathbf{z}|\mathbf{x})$. With samples from $q$ distribution, we can train $p$ by optimizing the variational lower bound:

$$\log p(\mathbf{x}) \geq \mathbb{E}_{\mathbf{z} \sim q(\mathbf{z}|\mathbf{x})} \log \frac{p(\mathbf{x}, \mathbf{z})}{q(\mathbf{z}|\mathbf{x})}. \tag{1}$$

Meanwhile, $q(\mathbf{z}|\mathbf{x})$ has to be optimized towards $p(\mathbf{z}|\mathbf{x})$ in order to keep the variational bound tight.

In the Wake-Sleep algorithm (Hinton et al., 1995; 2006), the wake phase corresponds to maximizing the objective in Eq. 1 with respect to the parameter of $p(\mathbf{x}, \mathbf{z})$ using samples from $q(\mathbf{z}|\mathbf{x})$ given a datapoint $\mathbf{x}$. In the sleep phase, a pair of samples $\mathbf{z}, \mathbf{x}$ is drawn from the generative distribution $p(\mathbf{x}, \mathbf{z})$ and then $q$ is trained to minimize the KL divergence $KL(p(\mathbf{z}|\mathbf{x}) \parallel q(\mathbf{z}|\mathbf{x}))$. This objective, however, is not theoretically sound as we should instead be minimizing its reverse: $KL(q(\mathbf{z}|\mathbf{x}) \parallel p(\mathbf{z}|\mathbf{x}))$.

Reweighted Wake-Sleep (RWS) (Bornschein & Bengio, 2014) brings two major improvements to the original Wake-Sleep algorithm. The first one is to reformulate the log-likelihood objective as an importance-weighted average and derive a tighter lower bound:

$$\log p(\mathbf{x}) = \log \mathbb{E}_{\mathbf{z}_i \sim q(\mathbf{z}|\mathbf{x})} \left[ \frac{1}{K} \sum_{i=1}^{K} \frac{p(\mathbf{x}, \mathbf{z}_i)}{q(\mathbf{z}_i|\mathbf{x})} \right] \geq \mathbb{E}_{\mathbf{z}_i \sim q(\mathbf{z}|\mathbf{x})} \left[ \log \frac{1}{K} \sum_{i=1}^{K} \frac{p(\mathbf{x}, \mathbf{z}_i)}{q(\mathbf{z}_i|\mathbf{x})} \right] = \mathcal{L}_K. \tag{2}$$

In the wake phase of RWS, parameters in $p$ are learned by optimizing the new lower bound defined in Eq. 2 (Burda et al., 2015). The second improvement is to add a wake phase for learning $q$. The wake phase for $q$ can be viewed as minimizing the KL divergence $KL(p(\mathbf{z}|\mathbf{x}) \parallel q(\mathbf{z}|\mathbf{x}))$ for a given datapoint $\mathbf{x}_{\text{data}}$ instead of $\mathbf{x}_{\text{sample}}$ as in the sleep phase. The authors empirically show that the new wake phase works better than the sleep phase in the original Wake-Sleep and works even better when combined with the sleep phase.

Although RWS tightens the bound and works well in practice, it still does not optimize a well-defined objective for inference network $q$. Mnih & Rezende (2016) propose a new method named VIMCO, which solves this problem. In VIMCO, both $p$ and $q$ are optimized jointly against the lower bound in Eq. 2. However, the gradient w.r.t parameters in $q$ will have high variance if we compute them naively. VIMCO algorithm utilizes the multiple samples to compose a baseline for each sample using the rest of samples (we refer readers to the original paper for more technical details). The author shows that VIMCO performs equally well as RWS when training SBNs on MNIST.

## 3 MODEL

Let us consider a latent variable model $p(\mathbf{x}) = \sum_{\mathbf{z}} p(\mathbf{x}|\mathbf{z})p(\mathbf{z})$ with the distribution $p(\mathbf{z})$ defined over the latent space. In addition, an inference network $q(\mathbf{z}|\mathbf{x})$ is used to approximate the intractable posterior distribution $p(\mathbf{z}|\mathbf{x})$. This fundamental formulation is shared by many deep generative models with latent variables, including deep belief networks (DBNs), and variational autoencoders (VAEs). Different realizations result in different architectures and corresponding learning algorithms. In our model, the prior distribution $p_\varphi(\mathbf{z})$ is multivariate Bernoulli modeled by a restricted Boltzmann machine (RBM) with a parameter vector $\varphi$. The approximate posterior distribution $q_\phi(\mathbf{z}|\mathbf{x})$ is multivariate Bernoulli with its mean modeled by a deep neural network $\phi$. The generative distribution $p_\theta(\mathbf{x}|\mathbf{z})$ is modeled by a deep neural network $\theta$ as well. Note that both networks are deterministic.

This model has several advantages. First, compared with VAEs, RBMs can handle both discrete and continuous latent variables. It also allows for a much richer family of latent distributions compared to simple factorized distributions as in vanilla VAEs. Although for VAEs, the posterior distribution is regularized towards a factorized Gaussian prior by minimizing KL divergence, in practice the KL divergence is never zero, especially when modeling complex datasets. Such discrepancy between the

prior and learned posterior can often damage the generative quality. The RBM approach, however, instead of pulling the posterior to some pre-defined prior, learns to mimic the posterior. During generation process, prior samples drawn by running the Markov chain defined by the RBM can often lead to images with higher visual quality than those drawn from vanilla VAEs.

Second, compared with SBNs and DBNs, only communication between inference and generative networks uses stochastic binary states. In this case, the inference and generative networks become fully differentiable so that multiple layers can be jointly optimized by back-propagation. This is radically different from SBNs and DBNs where each inference layer is trained to approximate the posterior distribution of a specific generative layer. Our framework can greatly increase the model capacity by allowing more complicated transformation between high dimensional input space and latent space. In addition, networks can exploit modern network design techniques, including convolution, pooling, dropout (Srivastava et al., 2014), or even ResNet (He et al., 2015) and DenseNet (Huang et al., 2016), in a very easy and straightforward way. Therefore, similar to VAEs, models under this framework can be scaled to handle more complex datasets compared to traditional SBNs and DBNs.

### 3.1 Pretraining with Autoencoder

Training a hybrid model is never a trivial task. Notably in our case, the encoder and decoder networks can be very deep and gradient cannot be propagated through stochastic states. In addition, RBMs are often more sensitive to training compared to feed-forward neural networks. Therefore, as in DBNs and DBMs, a clever pretraining algorithm that can help find a good weight initialization can be very beneficial.

To learn a good image prior, we jointly train parameters of the inference network $\phi$ and generative network $\theta$ as an autoencoder. To obtain a binary latent space, we use additive i.i.d uniform noise (Ballé et al., 2016) together with a modified hardtanh function to realize "soft-binarization".[1] This method can be described as the following function:

$$\mathcal{B}(\mathbf{z}) = f(\mathbf{z} + \mathcal{U}(-0.5, 0.5)), \text{ where } f(x) = \begin{cases} 0, x \leq 0 \\ x, 0 \leq x \leq 1 \\ 1, x \geq 1 \end{cases} \tag{3}$$

and $\mathbf{z} = \mathcal{E}(\mathbf{x})$ is the output of the encoder. This soft-binarization function will encourage the encoder to encode $\mathbf{x}$ into $([-\infty, -1] \cup [1, +\infty])^{|\mathbf{z}|}$ to maximize the information that follows through this bottleneck while allowing gradient descent methods to find such solution efficiently. To avoid overfitting, dropout can be applied after $\mathcal{B}$ function. The adoption of dropout in $\mathbf{z}$ space can also prevent the co-adaptation between latent codes, which makes it easier for RBMs to model.

In practice, we find that this pretraining procedure produces well binarized latent space on which RBMs can be successfully trained. Therefore, we can then pretrain the RBMs on $\mathbf{z}$ using contrastive divergence (Hinton, 2002) or persistent contrastive divergence (Tieleman, 2008). After pretraining, we remove $\mathcal{B}$ and append a sigmoid function $\sigma$ to the end of the encoder to convert it to the inference network, $i.e. \phi = \sigma \circ \mathcal{E}$. The decoder is then used to initialize the generator $\theta$.

### 3.2 Training

Since our model shares the fundamental formulation with many of the existing variational based models, we can modify the state-of-the-art learning algorithms to train it. The specific algorithms we are interested in are the reweighted wake-sleep (RWS) (Bornschein & Bengio, 2014) and VIMCO (Mnih & Rezende, 2016) which give the best results on SBN models and can handle discrete latent variables.

As reviewed in the background section, both RWS and VIMCO are importance sampling based methods, that need to compute weights $w_i = p(\mathbf{x}, \mathbf{z}_i)/q(\mathbf{x}|\mathbf{z}_i)$ for multiple samples $\mathbf{z}_i$ given input $\mathbf{x}$. These weights are then normalized as $\tilde{w}_i = w_i / (\sum_j w_j)$ to decide the contribution of each sample to the gradient estimator. In our model, the joint probability $p(\mathbf{x}, \mathbf{z})$ is intractable due to the partition

---

[1]Although real-valued data can be modeled by a Gaussian-Bernoulli RBM, it is often much harder to train and not the focus of this paper.

$\mathcal{Z}$ function introduced by RBM. However, it can be substituted by its unnormalized counterpart:

$$p^*(\mathbf{x}, \mathbf{z}) = \mathcal{Z}p(\mathbf{x}, \mathbf{z}) = e^{-\mathcal{F}(\mathbf{z})}p(\mathbf{x}|\mathbf{z}), \tag{4}$$

as the coefficient $\mathcal{Z}$ will be canceled during the weight normalization step. The $\mathcal{F}(\mathbf{z})$ is the free energy assigned to $\mathbf{z}$ by RBM, which can be computed analytically.

The RBM is also trained using multiple samples as part of the generative module. In both RWS and VIMCO, the gradient for RBM with parameter $\varphi$ is:

$$\frac{\partial}{\partial \varphi}\mathcal{L}_K \simeq \sum_{i=1}^{K} \tilde{w}_i \frac{\partial}{\partial \varphi} \log p_\varphi(\mathbf{z}_i) = -\sum_{i=1}^{K} \tilde{w}_i \frac{\partial \mathcal{F}(\mathbf{z}_i)}{\partial \varphi} + \mathbb{E}_{\mathbf{z}^- \sim \mathcal{M}}\left[\frac{\partial \mathcal{F}(\mathbf{z}^-)}{\partial \varphi}\right]. \tag{5}$$

The second term in Eq. 5 is the intractable model dependent term which needs to be estimated using samples from the RBM. The samples are obtained by running a persistent Markov chain as in persistent contrastive divergence (Tieleman, 2008).

There is one more modification for the sleep phase in RWS. The generative process now starts from a Markov chain defined by RBM instead of a direct draw from unconditional Bernoulli prior. This can be seen as the contrastive version of RWS. For completeness, we put the detail of Contrastive RWS and VIMCO in Appendix A.

### 3.3 EVALUATION

Quantitative evaluation of deep generative models is very crucial to measure and compare different probabilistic models. Fortunately, we can refer to a rich set of existing techniques for quantitative evaluations. One way to evaluate our model is to decompose the lower bound $\mathcal{L}_K$ in Eq. 1 as follows:

$$\mathcal{L}_K = \mathbb{E}_{\mathbf{z}_i \sim q(\mathbf{z}|\mathbf{x})}\left[\log \frac{1}{K}\sum_{i=1}^{K}\frac{p^*(\mathbf{x}, \mathbf{z}_i)}{q(\mathbf{z}_i|\mathbf{x})}\right] - \log \mathcal{Z} \tag{6}$$

and estimate partition function $\mathcal{Z}$ with Annealed Importance Sampling (AIS) (Salakhutdinov & Murray, 2008). This method is very efficient since we only need to estimate $\mathcal{Z}$ once no matter how large the $K$ is. However, since AIS gives an unbiased estimate of $\mathcal{Z}$, it on average tends to underestimate $\log \mathcal{Z}$ since $\log \mathcal{Z} = \log \mathbb{E}(\hat{\mathcal{Z}}) \geq \mathbb{E}(\log \hat{\mathcal{Z}})$ (Burda et al., 2014).

Another method that yields conservative estimates is Reverse AIS Estimator (RAISE) (Burda et al., 2014), which returns an unbiased estimate of $p(\mathbf{z})$. However, RAISE can be quite time consuming since it needs to run an independent chain for every $\mathbf{z}$. Therefore, we suggest to use AIS as main tool for evaluation during training and model comparison, since empirically AIS provides fairly accurate estimates, but also run RAISE as a safety check before reporting final results to avoid unrealistically high estimates of $\mathcal{L}_K$.

## 4 RELATED WORK

In Figure 1, we show the visualization of our model together with three closely related existing latent variable models, DBNs (Hinton et al., 2006), DEMs (Ngiam et al., 2011) and VAEs (Kingma & Welling, 2013).

The major difference between DBNs and our models is that every layer in the inference and generative networks in DBNs is stochastic. Such design drastically increases the difficulty in training and restrict the model from using modern deep convolutional architectures. Although convolution, combined with a sophisticated probabilistic pooling technique, has been applied to DBNs (Lee et al., 2009), the resulted convolutional DBNs is still difficult to train. It is also unclear how more recent techniques like residual connections (He et al., 2015) can be adapted for them. Our models, on the other hand, can integrate these techniques easily and learn deeper networks effectively.

The deep energy models (DEMs) by Ngiam et al. (2011) are previous attempts to use multiple deterministic layers to build deep generative models. In their setting, only the top-most layer, which resembles the hidden layer in an RBM, is stochastic. There is no explicit generator in the model and sampling is carried out through Hamiltonian Monte Carlo (HMC). In practice, we find that HMC

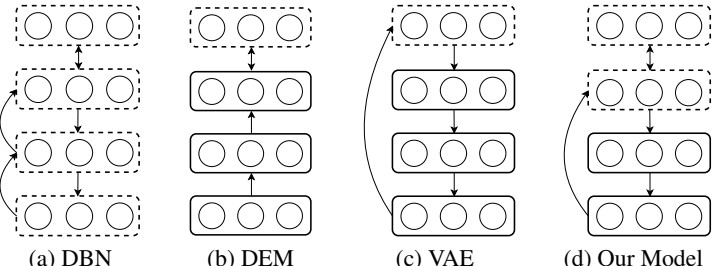

(a) DBN          (b) DEM          (c) VAE          (d) Our Model

Figure 1: Comparison between (**a**) deep belief networks, (**b**) deep energy models, (**c**) variational autoencoders and (**d**) our models. Dashed boxes denote stochastic layers and solid boxes denote deterministic layers. Bidirectional arrows denote undireicted connections. For simplicity, recognition networks in VAE and our model are represented by a single upward arrow.

samplers are too sensitive to hyper-parameters, making them extremely hard to use for sampling from deep convolutional networks. The generator solution in our model is simpler, more robust, and more scalable.

VAEs (Kingma & Welling, 2013) are modern deep generative models that have shown impressive success in a wide variety of applications (Yang et al., 2017; Pu et al., 2016). VAEs are directed graphical models that also consist of stochastic latent layers and deterministic deep neural networks. However, VAEs use factorized prior distributions, which can potentially limit the networks' modeling capacity by placing a strong restriction on the approximate posterior (Burda et al., 2015). There have been several works trying to resolve this issue by deriving more flexible posteriors (Jimenez Rezende & Mohamed, 2015; Kingma et al., 2016). The RBM in our model can represent more complex prior distributions by design, which can possibly lead to more powerful models.

PixelRNNs (van den Oord et al., 2016a) and GANs (Goodfellow et al., 2014) are two other popular generative models. PixelRNNs are fully observable models that use multiple layer of LSTMs to model images as a sequence of pixels. PixelRNN and its various variant PixelCNN (van den Oord et al., 2016b; Salimans et al., 2017; Gulrajani et al., 2016) exhibit excellent capacity on modeling local detail on images and are the state-of-the-art models in terms of density estimation. GANs simultaneously train a discriminator and a generator. The discriminator is trained to distinguish generated samples from real data while generator is trained to fool discriminator by generating realistic samples. GANs can generate visually appealing images but they are hard to evaluate quantitatively. Although several methods have been discussed recently (Theis et al., 2015; Wu et al., 2016), quantitative evaluation of GANs still remains a challenging problem.

## 5 EXPERIMENTS

We now describe our experimental results. Through a series of experiments we 1) quantitatively evaluate the importance of the pretraining step and compare the performance of our model trained with Contrastive RWS and VIMCO algorithms, 2) scale our model with ResNet (He et al., 2015) to approach the state-of-the-art result on MNIST, 3) scale our model to modeling images of natural scenes, and show that it performs comparable with its continuous counterparts in terms of density estimation, while being able to generate coherent samples. Please refer to Appendix C for details on the hyper-parameters and network architectures.

### 5.1 MNIST

We run our first set of experiments on the statically binarized MNIST dataset (Salakhutdinov & Murray, 2008; Larochelle & Murray, 2011). To model binary output, the generator $\theta$ computes the mean of the Bernoulli distribution $p_\theta(\mathbf{x}|\mathbf{z})$. We first train a simple model where both inference and generative networks are multi-layer perceptrons. The inference network contains five layers (784-200-200-100-100-200) and the generator contains the same layers in reverse order. We use ELU (Clevert et al., 2015) as our activation function. Note that the final layer in the inference network

| Model | NLL Test |
|---|---|
| DBN [1] | 84.55 |
| AR-SBN/SBN (RWS) [2] | 84.18 |
| IWAE [3] | 85.32 |
| Our Model (VIMCO, no pretrain) | 121.65 |
| Our Model (Contrastive RWS) | 84.33 |
| Our Model (VIMCO) | 83.69 (83.77) |

| Model | NLL Test |
|---|---|
| DRAW [1] | 80.97 |
| IAF VAE [2] | 79.88 |
| PixelRNN [3] | 79.20 |
| VLAE [4] | 79.03 |
| Gated PixelVAE [5] | 78.96 |
| Our ResNet Model | 79.58 (79.64) |

Table 1: Average test negative log-probabilities on MNIST. [1] Salakhutdinov & Murray (2008), [2] Bornschein & Bengio (2014), [3] Burda et al. (2015). Numbers of our model are computed with the AIS method in Section 3.3 while number in parenthesis is computed with the RAISE method.

Table 2: Average test negative log-probabilities on MNIST. [1] Gregor et al. (2015), [2] Kingma et al. (2016), [3] van den Oord et al. (2016a) [4] Chen et al. (2016), [5] Gulrajani et al. (2016).

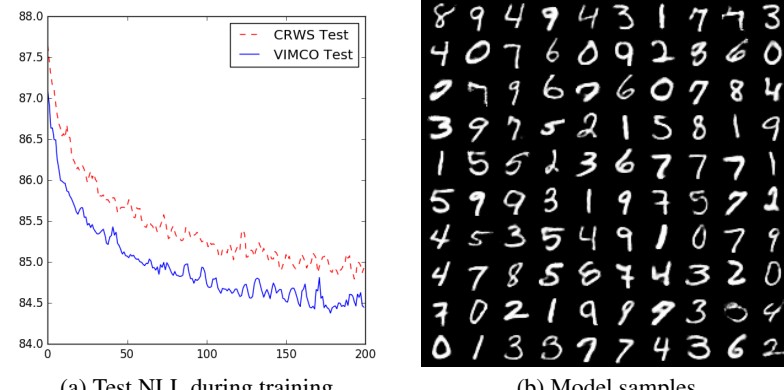

(a) Test NLL during training

(b) Model samples

Figure 2: **Left:** Negative log-likelihood on MNIST test set during training with Contrastive RWS and VIMCO. **Right:** Samples generated by running Gibbs sampling in RBM for 1000 steps and passing generated $\mathbf{z}$ through generator $\theta$, no further sampling in pixel space.

is normally larger since it is supposed to transmit only binary information. The RBM has 200 visible units $\mathbf{z}$ and 400 hidden units $\mathbf{h}$. The model is first pretrained and then trained with Contrastive RWS or VIMCO using 50 samples per data point. The learning curves of the first 200 epochs for models trained with both methods are shown in Figure 2a.

To evaluate our model, we use the AIS method following Eq.6 in Section 3.3 with $K = 5e4$. $\mathcal{Z}$ is estimated by running 5000 AIS chains with $1e5$ intermediate distributions. Table 1 shows performance of our fully connected model together with several previous works that use a similar network size. From the table and the learning curves in Figure 2a, we can see that our model trained with VIMCO objective performs better compared to training with Contrastive RWS. The superiority of VIMCO over Contrastive RWS is consistent during our experiments with various network configurations. In addition, we need to carefully tune the learning rate for inference and generative network separately to make Contrastive RWS work well on our model, which may be caused by the fact that wake-sleep algorithms are not optimizing a well defined objective.

To emphasize the importance of the pretraining algorithm, we also train the model with VIMCO directly starting from random initialization and it performs significantly worse. This result shows that the pretraining stage is very effective and is also necessary to make our model work. We also evaluate the best model with RAISE method to make sure that the result is not over-optimistic. For RAISE, we use fewer samples ($K = 5e3$) due to its high computation cost. The RAISE result is shown in the parenthesis in Table 1. The two estimators agree closely with each other, indicating that the results are accurate. Finally, we show samples from our model trained with VIMCO in Figure 2b.

Comparing with other methods in Table 1, our model clearly outperforms previous models that use multiple stochastic layers with or without RBM. The improvement indicates that using continuous

| Model | NLL Train | NLL Test |
|---|---|---|
| ResNet VAE with IAF [1] | | 3.11 |
| DenseNet VLAE [2] | | 2.95 |
| PixelCNN++ [3] | | 2.92 |
| IWAE | 4.45 | 4.54 |
| Our Model | 4.73 | 4.84 |

Table 3: Average test negative log-probabilities on CIFAR10 in bits/dim. [1] Kingma et al. (2016), [2] Chen et al. (2016), [3] Salimans et al. (2017). Numbers of our model are computed with the AIS method.

deep networks can indeed result in better performance in terms of density estimation. Notably, our model also outperforms IWAE (Burda et al., 2015), which in principle can be seen as the continuous counterpart of our model without an RBM prior. To fully test the capacity of our framework, we train a deep convolutional model with ResNet (He et al., 2015) blocks. The result is shown in Table 2. Our model surpasses the previous best models that use purely VAEs (Gregor et al., 2015; Kingma et al., 2016) and is only slightly behind the state-of-the-art models that use PixelRNN (van den Oord et al., 2016a) or VAEs combined with PixelCNNs (Chen et al., 2016; Gulrajani et al., 2016). In principle, PixelCNN can also be integrated into our framework as decoder, but we will leave this for future work.

## 5.2 CIFAR10

CIFAR10 has been a challenging benchmark for generative modeling. To model real value pixel data, we set the generative distribution $p_\theta(\mathbf{x}|\mathbf{z})$ to be discretized logistic mixture following Salimans et al. (2017). In the pretraining stage, the objective is to minimize the negative log-likelihood. The marginal distribution of the encoded $\mathbf{z}$ space and the reconstruction of test images are shown in Figure 5 in Appendix B. We note that the pretrained autoencoder preserves enough information while converting high dimensional real value data to binary. This transformation makes it possible apply simple models like RBM to challenging tasks such as modeling CIFAR10 images. For the network architecture, we use ResNets (He et al., 2015) for inference and generative networks. The latent space has 1024 dimensions and is modeled by RBM with 2048 hidden units. Similar to what we have discovered during the MNIST experiments, we find that VIMCO is more effective and robust to hyperparameters than Contrastive RWS. Therefore, the model is trained using VIMCO with 10 samples per data point.

For quantitative and qualitative comparisons under controlled variates, we train an IWAE (Burda et al., 2015) with roughly the same networks and the same amount of posterior samples per data point. The quantitative results are shown in Table 3 and samples from both models are shown in Figure 3a and Figure 4a. Here, our model performs slightly worse than IWAE in terms of density estimation, but the samples from our model have much higher visual quality. Note that results from both models are far behind those from state-of-the-art models (Salimans et al., 2017). To achieve significantly better results for VAE family models on CIFAR10, we often need to use more complicated networks with multiple sets of latent variables (Kingma et al., 2016) or use autoregressive decoders for output distribution (Chen et al., 2016) or both (Gulrajani et al., 2016). In this work, however, we keep our models simple to focus on the learning procedure.

To facilitate visual comparison, we also reproduce samples from a popular GAN model (Arjovsky et al., 2017) in Figure 4b. Samples from our model look natural, coherent but blurry, while samples from WGAN look clear, detailed but distorted. We admit that with many advanced techniques (Salimans et al., 2016; Arora et al., 2017; Dai et al., 2017), GANs still produce the highest quality images. However, our model has the advantage that it can be properly evaluated as a density model. Additionally, the flexibility of our framework could also accommodate potential future improvements.

## 5.3 IMAGENET64

We next use the $64 \times 64$ ImageNet (van den Oord et al., 2016a) to test the scalability of our model. Figure 3b, shows samples generated by our model. Although samples are far from being realistic

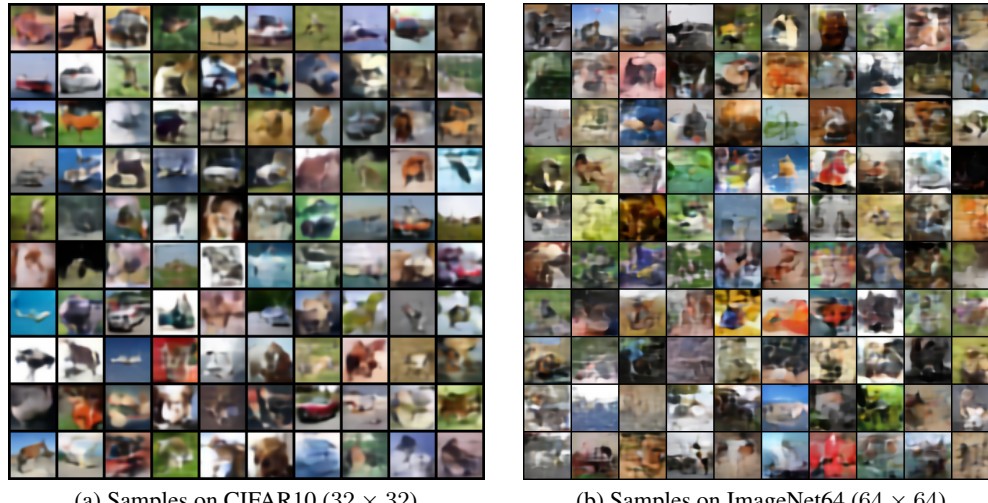

(a) Samples on CIFAR10 (32 × 32)  (b) Samples on ImageNet64 (64 × 64)

Figure 3: Samples generated by our model trained on CIFAR10 (left) and ImageNet64 (right).

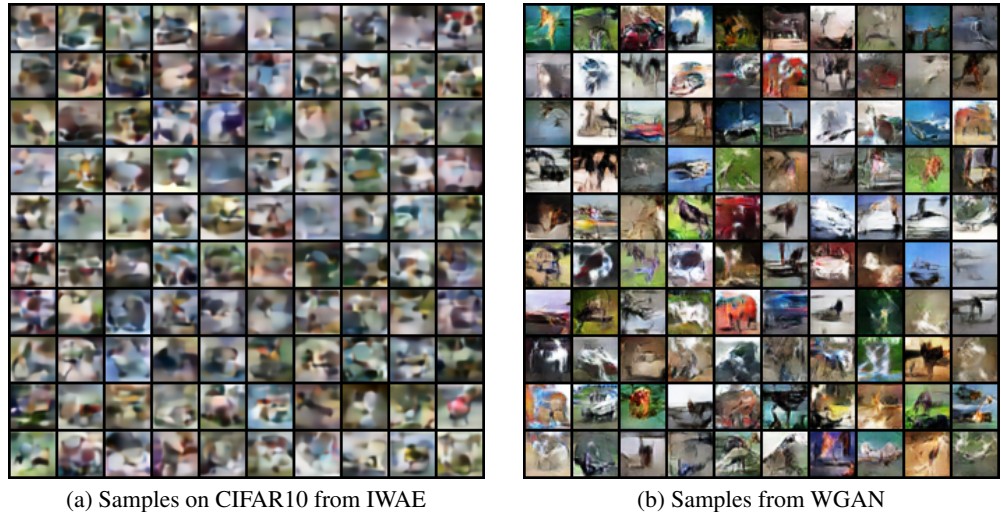

(a) Samples on CIFAR10 from IWAE  (b) Samples from WGAN

Figure 4: Samples generated by IWAE (left) and WGAN (right) trained on CIFAR10

and have strong artifacts, many of them look coherent and exhibit a clear concept of foreground and background, which demonstrates that our method has a strong potential to model high resolution images. The density estimation performance of this model is 4.92 bits/dim.

# 6 CONCLUSION

In this paper we presented a novel framework for constructing deep generative models with RBM priors and develop efficient learning algorithms to train such models. Our models can generate appealing images of natural scenes, even in the large-scale setting, and, more importantly, can be evaluated quantitatively. There are also several interesting directions for further extensions. For example, more expressive priors, such as those based on deep Boltzmann machines (Salakhutdinov & Hinton, 2009), can be used in place of RBMs, while autoregressive (Gregor et al., 2013) or recurrent networks (van den Oord et al., 2016a) can be used for inference and generative networks.

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

# A DETAILS ON CONTRASTIVE REWEIGHTED WAKE-SLEEP AND VIMCO

## A.1 CONTRASTIVE REWEIGHTED WAKE-SLEEP

---

**Algorithm 1** Contrastive Reweighted Wake-Sleep for a single observation

---

1: Sample $\mathbf{x}$ from training distribution.
2: **for** $i = 1$ to $K$ **do**
3:    Sample $\mathbf{z}_i$ from $q_\phi(\mathbf{z}|\mathbf{x})$.
4: **end for**
5: Compute unnormalized weights $w_i = \frac{p_\theta^*(\mathbf{x}, \mathbf{z}_i)}{q_\phi(\mathbf{z}_i|\mathbf{x})}$
6: Normalize the weights $\tilde{w}_i = \frac{w_i}{\sum_{i'} w_{i'}}$
7: Sample $\{\mathbf{z}^-\}$ from $M$ CD/PCD chains and pass through generator $\theta$ to obtain $\{\mathbf{x}^-\}$
8: Wake update for generative network $\theta$ with gradient:

$$\sum_{i=1}^{K} \tilde{w}_i \nabla_\theta \log p_\theta(\mathbf{x}, \mathbf{z}_i)$$

9: Wake and sleep updates for inference network $\phi$ with gradient:

$$\sum_{i=1}^{K} \tilde{w}_i \nabla_\phi \log q_\phi(\mathbf{z}^{(k)}|\mathbf{x}) + \frac{1}{M} \sum_{j=1}^{M} \nabla_\phi \log q_\phi(\mathbf{z}_j^-|\mathbf{x}_j^-)$$

10: Update RBM $\varphi$ with gradient:

$$-\sum_{i=1}^{K} \tilde{w}_i \nabla_\varphi \mathcal{F}_\varphi(\mathbf{z}_i) + \frac{1}{M} \sum_{j=1}^{M} \nabla_\varphi \mathcal{F}_\varphi(\mathbf{z}_j^-)$$

---

## A.2 VIMCO

---

**Algorithm 2** VIMCO for a single observation

---

1: Sample $\mathbf{x}$ from training distribution.
2: **for** $i = 1$ to $K$ **do**
3:    Sample $\mathbf{z}_i$ from $q_\phi(\mathbf{z}|\mathbf{x})$
4: **end for**
5: Compute unnormalized weights $w_i = \frac{p_\theta^*(\mathbf{x}, \mathbf{z}_i)}{q_\phi(\mathbf{z}_i|\mathbf{x})}$
6: Compute multi-sample variational bound: $\mathcal{L}_K = \log \frac{1}{K} \sum_{i=1}^{K} w_i$
7: **for** $i = 1$ to $K$ **do**
8:    Compute geometric mean of the rest of samples: $w_{-i} = \left( \prod_{j \neq i} w_j \right)^{\frac{1}{K-1}}$
9:    Compute the baseline learning signal: $\mathcal{L}_{-i} = \log \frac{1}{K} \left( w_{-i} + \sum_{j \neq i} w_j \right)$
10: **end for**
11: Normalize the weights $\tilde{w}_i = \frac{w_i}{\sum_{i'} w_{i'}}$
12: Sample $\{\mathbf{z}^-\}$ from $M$ CD/PCD chains and pass through generator $\theta$ to obtain $\{\mathbf{x}^-\}$
13: Update generative network $\theta$ with gradient: $\sum_{i=1}^{K} \tilde{w}_i \nabla_\theta \log p_\theta(\mathbf{x}, \mathbf{z}_i)$
14: Update inference network $\phi$ with gradient:

$$\sum_{i=1}^{K} (\mathcal{L}_K - \mathcal{L}_{-i} - \tilde{w}_i) \nabla_\phi \log q_\phi(\mathbf{z}_i|\mathbf{x})$$

15: Update RBM $\varphi$ with gradient:

$$-\sum_{i=1}^{K} \tilde{w}_i \nabla_\varphi \mathcal{F}_\varphi(\mathbf{z}_i) + \frac{1}{M} \sum_{j=1}^{M} \nabla_\varphi \mathcal{F}_\varphi(\mathbf{z}_j^-)$$

---

# B    QUALITATIVE EVALUATION OF PRETRAINED MODEL ON CIFAR10

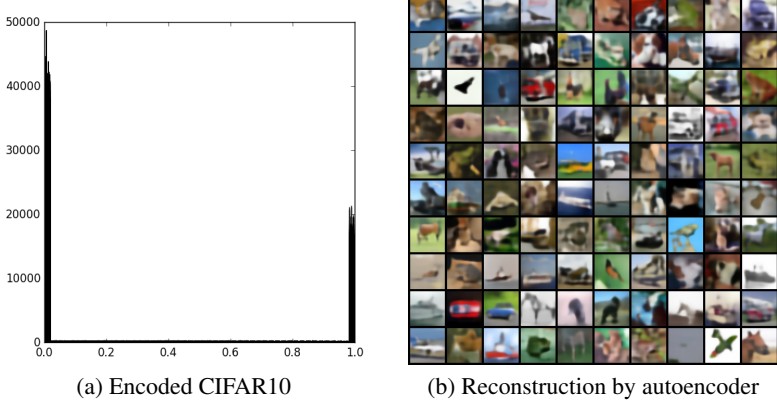

(a) Encoded CIFAR10                    (b) Reconstruction by autoencoder

Figure 5: **Left:** Marginal distribution of $\mathbf{z}$ in the encoded CIFAR10. **Right:** Reconstruction of test images. These are expected value of the output distribution without further sampling.

# C    EXPERIMENTAL SETUP

In this section, we describe the training details and network configurations for experiments in Section 5. Code will be released as well.

The general training procedure is as follows. We first pretrain the inference and generative networks as autoencoder by maximizing log-likelihood on training data. Then we pretrain RBM with contrastive divergence starting from 1 step (CD1) and gradually increase to 25 steps (CD25). This training method has been previously used to produce the best RBM on MNIST dataset (Salakhutdinov & Murray, 2008). We additionally train the RBM using persistent contrastive divergence with 25 steps (PCD25) or more. Finally, we train all three components jointly with Contrastive RWS or VIMCO. In Contrastive RWS and VIMCO, samples from RBM are drawn from a persistent chain. We use SGD with learning rate decay for learning RBMs and Adam or Adamax (Kingma & Ba, 2014) elsewhere.

We experiment with three activation functions ReLU, LeakyReLU and ELU (Clevert et al., 2015), and find out that ELU performs slightly better. Inspired by (Kingma et al., 2016), we use weight normalization (Salimans & Kingma, 2016) in deep ResNet models as we find that it works better than batch normalization for our model as well.

In MNIST experiments, the shallow fully connected model uses an inference network with five layers (784-200-200-100-100-200) and a generative network with the same layers in reversed order. The RBM has 200 visible units $\mathbf{z}$ and 400 hidden units $\mathbf{h}$. For the deep ResNet model, the inference network uses three basic pre-activation (He et al., 2016) residual blocks with 25, 50, 50 feature maps. Each block uses kernel size 3 and is repeated twice with stride 2 and 1 respectively. After residual blocks, there is a fully connected layer with 200 neurons. The RBM has 200 visible units and 400 hidden units. The generative network uses the same blocks but with stride one. We upsample the feature map with nearest neighbour interpolation by a factor of 2 before feeding it into each block and shortcut to avoid checkerboard artifact (Odena et al., 2016).

In CIFAR10 and ImageNet64 experiments, the output distribution $p_\theta$ is a discretized mixture of 10 logistic distributions (Salimans et al., 2017). The network for CIFAR10 uses 4 residual blocks with 64, 128, 192, 256 feature maps. Each block is repeated twice as in MNIST. There is no fully connected layer in this model and final feature map ($256 \times 2 \times 2$) is flattened to a 1024 dimensional vector. The RBM has 1024 visible units and 2048 hidden units. The network for ImageNet64 uses 5 residual blocks with 64, 128, 128, 256, 256 feature maps. Each block uses stride 2 and is only repeated once. The RBM is the same as the one in CIFAR10.

