# OpenReview forum: "Learning Deep Generative Models With Discrete Latent Variables"
_ICLR.cc/2018/Conference — Reject_

### Official Review · AnonReviewer2 · 2017-11-18
**The paper proposes to augment a variational auto encoder (VAE) with an binary restricted Boltzmann machine (RBM) in the role of the prior of the generative model. Clarity could be improved in some aspects and the advantages of the proposed model compared to existing ones did not become totally clear to me.**

**Rating:** 4
**Confidence:** 4

**Review:**

Summary of the paper:
The paper proposes to augment a variational auto encoder (VAE) with an binary restricted Boltzmann machine (RBM) in the role of the prior of the generative model. To yield a good initialisation of the parameters of the RBM and the inference network a special pertaining procedure is introduced. The model produces competitive Likelihood results on MNIST and was further tested on CIFAR 10.

Clarity and quality:

1. From the description of the pertaining procedure and the appendix B I got the impression that the inference network maps into [0,1] and not into {0,1}.  Does it mean, you are not really considering binary latent variables (making the RBM model the values in [0,1] by its probability p(z|h))?

2. on page 2:
RWS...."derive a tighter lower bound": Where does the "tighter" refer to?

3. "multivariate Bernoulli modeled by an RBM": Note,  while in a multivariate Bernoulli the binary variables would be independent from each others, this is usually not the case for the visible variables of RBMs (only in the conditional distribution given the state of the hidden variables).

4. The notation could be improved, e.g.:
-x_data and x_sample are not explained
- M is not defined in equation 5.

5. "this training method has been previously used to produce the best results on MNIST" Note, that parallel tempering often leads to better results when training RBMs (see http://proceedings.mlr.press/v9/desjardins10a/desjardins10a.pdf) . Furthermore, centred RBMs are also get better results than vanilla RBMs (see: http://jmlr.org/papers/v17/14-237.html).

Originality and significance:
As already mentioned in a comment on open-review the current version of the paper misses to mention one very related work: "discrete variational auto encoders". Also "bidirectional Helmholtz machines" could be mentioned as generative model with discrete latent variables.  The results for both should also be reported in Table 1 (discrete VAEs: 81,01, BiHMs: 84,3).

From the motivation the advantages of the model did not become very clear to me. Main advantage seems to be the good likelihood result on MNIST (but likelihood does not improve compared to IWAE on CIFAR 10 for example). However, using an RBM as prior has the disadvantage that sampling from the generative model requires running a Markov chain now while having a solely directed generative model allows for fast sampling.

Experiments show good likelihood results on MNIST. Best results are obtained when using a ResNet decoder. I wondered how much a standard VAE is improved by using such a powerful decoder. Reporting this, would allow to understand, how much is gained from using a RBM for learning the prior.

Minor comments:
page 1:
"debut of variational auto encoder (VAE) and reparametrization trick" -> debut of variational auto encoders (VAE) and the reparametrization trick",
page 2:
"with respect to the parameter of  p(x,z)" -> "with respect to the parameters of  p(x,z)"
"parameters in p" -> "parameters of p"
"is multivariate Bernoulli" ->  "is a multivariate Bernoulli"
"we compute them" -> "we compute it"
page 3:
"help find a good" ->  "help to find a good"
page 7:
"possible apply" -> "possible to apply"

---

### Official Review · AnonReviewer3 · 2017-11-22
**OK, but fairly incremental and results underwhelming**

**Rating:** 5
**Confidence:** 4

**Review:**

While I acknowledge that training generative models with binary latent variables is hard, I'm not sure this paper really makes valuable progress in this direction. The only results that seem promising are those on binarized MNIST, for the non-convolutional architecture, and this setting isn't particularly exciting. All other experiments seem to suggest that the proposed model/algorithm is behind the state of the art. Moreover, the proposed approach is fairly incremental, compared to existing work on RWS, VIMCO, etc.

So while this work seem to have been seriously and thoughtfully executed, I think it falls short of the ICLR acceptance bar.

---

### Official Review · AnonReviewer1 · 2017-11-27
**Interesting work**

**Rating:** 4
**Confidence:** 4

**Review:**

Interesting work, but I’m not convinced by the arguments nor by the experiments. Similar models have been trained before; it’s not clear that the proposed pretraining procedure is a practical step forwards. And quite some decisions seem ad-hoc and not principled.

Nevertheless, interesting work for everyone interested in RBMs as priors for “binary VAEs”.

---

### Public Comment · (anonymous) · 2017-10-31
**A more informative title would be helpful**

What were your reasons for choosing such a general title? It would be understandable if this paper were the first work in this area or if it provided some sort of unifying view of prior of work on such models (DBNs, DBMs, SBNs etc.), but it is not the case.

It would also be good to discuss how the proposed model is related to Discrete VAEs, which also combine an RBM with a directed mapping.

---

> ### Author Response · Authors · 2017-11-11
> **Comparison with Discrete VAE**
>
> Thank you for pointing out this paper! We will include the discussion on Discrete VAE in future revisions.
>
> Although both Discrete VAE and our model have discrete latent variables, the detailed architectures are different. In Discrete VAE, the RBM (or bipartite Boltzmann machine) is fully hidden(z) and connects to the encoder-decoder through a set of continuous smoothing variables. Our model uses a latent(z)-hidden(h) RBM. The visible layer of RBM directly connects to encoder-decoder and they only exchange discrete states.
>
> These two papers also have different focuses. In Discrete VAE, they introduce a method to project posterior and prior into a continuous space so that the discrete variables can be integrated out. This can be seen as “reparameterizing” discrete into continuous to make the autoencoder term fully differentiable. In our paper, we try to answer whether it is possible to train a DBN-inspired model without any reparameterization but with proper learning procedure. This results in a conceptually straightforward model that actually outperforms the Discrete VAE on MNIST (79.58 VS 81.01). We also study how well our method scales to real images, which is not mentioned in many previous works using discrete latent variables.

---

### Comment · AnonReviewer1 · 2017-11-27
**Interesting work**

Interesting work!

Just to clarify: You pretrain the encoder/decoder pair and pretrain a RBM on their latent representation? And then during joint training (section 3.2) you block direct gradient flow (disable the soft-binarization) and use VIMCO?

I’m not convinced I follow the second part of the argument in 3.0: Training SBNs or DBNs with e.g. Gumbel-softmax based methods would allow gradients to flow in a very similar fashion. Doesn’t the “gradient flow” depend on the training method / gradient estimator (in your case soft-binarization) rather than this models structure?

From this perspective your model is directly comparable to Discrete VAEs, isn’t it? Where Discrete VAEs introduced a different reparam. based training method.

Have you tried alternative pretraining methods? E.g. using Gumbel-softmax based instead eqn. (3), or VIMCO trained factorial top later SBN? Do we have any idea why joint training might be so hard?

The title seems very broad - large parts of the paper propose and evaluate a pretraining procedure for a specific two layer DBM architecture.

I’m curious: What are typical log Z estimate for your models in table 1?

---

> ### Author Response · Authors · 2017-11-29
> **Clarifications and comments**
>
> Thank you for your comments
>
> >>> Just to clarify: You pretrain the encoder/decoder pair and pretrain a RBM on their latent representation? And then during joint training (section 3.2) you block direct gradient flow (disable the soft-binarization) and use VIMCO?
>
> Yes. The direct gradient flow is blocked during joint training with VIMCO and Contrastive RWS.
>
>
> >>> I’m not convinced I follow the second part of the argument in 3.0: Training SBNs or DBNs with e.g. Gumbel-softmax based methods would allow gradients to flow in a very similar fashion. Doesn’t the “gradient flow” depend on the training method / gradient estimator (in your case soft-binarization) rather than this models structure?
>
> The difference between our model and SBNs or DBNs is better illustrated in Figure 1. In DBN and SBN, every layer is stochastic and defines its own generative distribution p(z_i+1|z_i). There are N stochastic layers and thus N intermediate generative distributions. Correspondingly, there are N approximate inference distributions (as denoted by multiple upward arrows). The i-th inference layer is trained to approximate the posterior distribution of i-th generative layer. The learning signal for each layer depends locally on the input output states, but not on the gradient information that is propagated from deeper layers. In our model, the decoder and encoder are deterministic and continuous. They define one pair of p(x|z) and q(z|x) and there are no intermediate stochastic layers. Thus multiple layers in encoder/decoder are trained with gradients that flow freely within encoder/decoder.
>
> If we train DBN or SBN with Gumbel-softmax based methods, the stochastic nodes in each layer are replaced with continuous approximations. In that case the gradient flow between layers is possible but not as freely as in our model because still in DBN or SBN, both the encoder and decoder would still be represented as stochastic layers. Since soft-binarization is only used during pretraining, our model does not contain any continuous relaxation during joint training (our encoder and decoder are continuous and deterministic), which is a major difference from continuous relaxation based methods.
>
>
> >>> From this perspective your model is directly comparable to Discrete VAEs, isn’t it? Where Discrete VAEs introduced a different reparam. based training method.
>
> Both Discrete VAEs and our model utilize deep continuous encoder/decoder for stronger representation power. Discrete VAEs have a layer of continuous latent variables between bipartite Boltzmann machine prior and encoder/decoder. They project posterior and prior into a continuous space to make the "autoencoder" part of the loss be fully differentiable.
>
> In our method, the "autoencoder" is fully differentiable during pretraining. In joint training, encoder and decoder only transmit stochastic states and thus block the gradient flow between them. Another difference in architecture is that Discrete VAEs use bipartite BM with both parts connected to the rest of the model while our models use 2 layer BM (RBM) with first layer connected to the latent space z and the second being fully hidden. In terms of density estimation, our model outperforms Discrete VAE on MNIST(79.58 vs. 81.01, as reported by [1]). We will clarify this point and add additional experimental results comparing our model to Discrete VAEs.
>
>
> >>> Have you tried alternative pretraining methods? E.g. using Gumbel-softmax based instead eqn. (3), or VIMCO trained factorial top layer SBN? Do we have any idea why joint training might be so hard?
>
> We tried using soft-binarization with Gaussian noise but that introduces strong artifacts in generated images. Pretraining with Gumbel-softmax based methods is an interesting idea. In their original paper [2][3], the methods are tested on models with one or two layers. Thus it is hard to conclude immediately whether that will work as effectively on deep networks. But that will be worth exploring. Pretraining with SBN is more tricky. Note that it still does not solve the problem that deep generative models are very hard to train from scratch with REINFORCE style algorithms and discrete latent variables. Pretraining with factorial prior also adds stronger constrain on the approximate posterior compared with the RBM prior.
>
>
> >>> The title seems very broad - large parts of the paper propose and evaluate a pretraining procedure for a specific two layer DBM architecture.
>
> Thank you for pointing this out, we will make the title be more focused.
>
>
> >>> I’m curious: What are typical log Z estimate for your models in table 1?
>
> Between 265 and 270.
>
>
> [1] Discrete Variational Autoencoders
> [2] Categorical Reparameterization with Gumbel-Softmax
> [3] The Concrete Distribution: A Continuous Relaxation of Discrete Random Variables

---

### Author Response · Authors · 2017-12-23
**Response and Clarification**

We thank reviewers for their valuable feedback. There are several points we would like to clarify.

Motivation: The motivation of the work is to study the effect of a learnable prior for generative models with latent variables. The learnable prior can potentially take forms of any graphical model, while here we focus on using RBM due to its simplicity. This strength over vanilla VAEs has been shown on CIFAR10 and ImageNet64 experiments. In addition, these models can still be quantitatively evaluated in terms of density estimation (as we demonstrate in our work), which is a benefit compared top GAN-like models since they are relatively hard to evaluate quantitatively.

Quantitative performance: Our model performs well on MNIST. The ResNet model uses a similar network as IAF-VAE, which was previously shown to achieve state-of-the-art results using deep convnets (cited in Table 2 and mentioned in the main text). IAF-VAE was about 2 nats better than vanilla VAE using the same ResNet architecture. Thus, the fact that our model performs slightly better than IAF-VAE shows that the learned RBM prior has some merits. The current state-of-the-art models use PixelCNN/PixelRNN-based decoders, which by themselves are very strong density estimation models. PixelCNN/PixelRNN-based decoders can potentially be integrated into our framework as well.

Regarding CIFAR10: Compressing real valued images into binary is a hard problem. In fact, if we increase the dimension of the latent space from 1024 to 2048 and use a 2048-4096 RBM, the performance of our model can be substantially improved, achieving Test NLL of 4.54 bits/dim, the same as that of IWAE.

We focus on comparing our results with IWAE as it represents a strong baseline that uses discrete latent variables. We also compared with models that use unconditional Bernoulli prior, but these models performed much worse, compared to both, IWAE and our model, in terms of both density estimation and generated samples.


Clarifications for Reviewer 2

Thank you for your detailed feedback.

1. Regarding inference network mapping into [0,1] and not into {0,1}:

As mentioned at the end of the pretraining section, the “soft-binarization” is removed after pertaining and a sigmoid layer is added to produce valid probabilities. During joint training, we only use samples {0, 1}.

2. Regarding tightness of the lower-bound:

Lower bound in Eq.2 is tighter than a single sample bound in Eq.1 (as was originally derived in IWAE paper). We will change the text to make this point clear.

3. Regarding "multivariate Bernoulli modeled by an RBM":

Thanks for pointing this out. Yes the use of multivariate Bernoulli is not rigorous here. We will change this in revision.

4. Improving notation:

Thanks for catching typos. The x_data refers to data points from training set while x_sample is samples from model distribution. M refers to the model distribution. We will check other notations as well.

5. Regarding parallel tempering/centred RBMs:

Thanks for pointing this out. Here we mean that this method has produced good RBM results in practice on MNIST in terms of density estimation, and has been widely used in practice (in addition to Persistent CD) We will correct this.

---

### Decision · Program_Chairs · 2018-01-29
**ICLR 2018 Conference Acceptance Decision**

**Decision:**

Reject

**Comment:**

The reviewers agreed that while this is a well-written paper, it is low on novelty and does not make a substantial enough contribution. They also pointed out that although the reported MNIST results are highly competitive, possibly due to the use of a powerful ResNet decoder, the CIFAR10/ImageNet results are underwhelming.